# The Past, Present, and Future of the Hemlock Woolly Adelgid (*Adelges tsugae*) and Its Ecological Interactions with Eastern Hemlock (*Tsuga canadensis*) Forests

**DOI:** 10.3390/insects9040172

**Published:** 2018-11-23

**Authors:** Aaron M. Ellison, David A. Orwig, Matthew C. Fitzpatrick, Evan L. Preisser

**Affiliations:** 1Harvard Forest, Harvard University, 324 North Main Street, Petersham, MA 01366, USA; orwig@fas.harvard.edu; 2Appalachian Laboratory, Center for Environmental Science, University of Maryland, 301 Braddock Road, Frostburg, MD 21532, USA; mfitzpatrick@umces.edu; 3Department of Biological Sciences, University of Rhode Island, Kingston, RI 02881, USA; preisser@uri.edu

**Keywords:** carbon flux, carbon stocks, dispersal, ecological forecasting, ecosystem dynamics, forest, invasive species, mortality, spread

## Abstract

The nonnative hemlock woolly adelgid is steadily killing eastern hemlock trees in many parts of eastern North America. We summarize impacts of the adelgid on these forest foundation species; review previous models and analyses of adelgid spread dynamics; and examine how previous forecasts of adelgid spread and ecosystem dynamics compare with current conditions. The adelgid has reset successional sequences, homogenized biological diversity at landscape scales, altered hydrological dynamics, and changed forest stands from carbon sinks into carbon sources. A new model better predicts spread of the adelgid in the south and west of the range of hemlock, but still under-predicts its spread in the north and east. Whether these underpredictions result from inadequately modeling accelerating climate change or accounting for people inadvertently moving the adelgid into new locales needs further study. Ecosystem models of adelgid-driven hemlock dynamics have consistently forecast that forest carbon stocks will be little affected by the shift from hemlock to early-successional mixed hardwood stands, but these forecasts have assumed that the intermediate stages will remain carbon sinks. New forecasting models of adelgid-driven hemlock decline should account for observed abrupt changes in carbon flux and ongoing and accelerating human-driven land-use and climatic changes.

## 1. Introduction

Eastern hemlock (*Tsuga canadensis* (L.) Carr.; Pinales: Pinaceae) is a long-lived foundation species (sensu [1,2]) that creates important structural diversity and habitat for a wide variety of vertebrate and arthropod species [3,4,5,6,7,8,9,10,11,12] and modulates core ecosystem processes such as energy and nutrient flows and water balance [11,13,14,15,16,17,18,19,20,21,22,23]. For nearly 10 millennia, *T. canadensis* has been a significant component of eastern North American forests [24]. For substantial intervals during this period, eastern hemlock pollen accounted for >60% of the grains in sediment cores, and in southern New England its abundance relative to other taxa likely exceeded 50% [25]. A regionally warmer and drier climate, perhaps interacting with the geographically coincident outbreak of a native insect (the hemlock looper *Lambdina fiscellaria* Guenée; Lepidoptera: Geometridae), led to an abrupt, range-wide decline of eastern hemlock ca. 5500 ka (thousands of calibrated radiocarbon years before 1950 C.E.); [26,27,28,29]. It took at least 1000 years for eastern hemlock to return to its pre-decline abundance in eastern North American forests.

*Tsuga canadensis* grows throughout eastern North America, from northern Georgia (USA) to southern Canada, and then west into Michigan and Wisconsin (Figure 1). Its total range exceeds 10,000 km^2^, and in many parts of its range, *T. canadensis* can account for the majority of individuals and total basal area in a given stand [30,31]. The U.S. National Vegetation Classification includes *T. canadensis* either as the dominant component or a significant, common component of more forest associations (14) than any other tree species [32]. In many parts of its range in eastern North America, the abundance of *T. canadensis* is declining because of a small, nonnative insect, the hemlock woolly adelgid (*Adelges tsugae* Annand; Hemiptera: Adelgidae). Here, we summarize impacts of the adelgid on eastern hemlock; review previous models and analyses of adelgid spread dynamics that estimated the spread of the adelgid and its impact on eastern hemlock forest ecosystems; and examine how those forecasts compare with current conditions. We use lessons learned from these studies to develop forecasts of future dynamics of the hemlock-hemlock woolly adelgid system.

## 2. The Hemlock—Hemlock Woolly Adelgid System

### 2.1. The Hemlock Woolly Adelgid

*Adelges tsugae* (henceforth, the “adelgid”) is a small (≈1.5 mm) insect that was first detected in the early 1950s in eastern North America near Richmond, Virginia; the detected lineage originated in Japan in [34,35]. In the subsequent six decades, the adelgid has established populations in 19 states—from Georgia to southern Maine and west into Michigan (Figure 1)—and in southwestern Nova Scotia (Canada) [36], where it threatens both eastern hemlock and the narrowly endemic Carolina hemlock (*T. caroliniana* Engelm.) (henceforth, “hemlock” refers to either or both species). The adelgid has two generations per year [37], and it has expanded its range rapidly since the 1980s [38]. Feeding by both generations of the adelgid on new needles leads to progressive needle loss, often first observed in the lower and central portions of tree crowns, then on interior branches and exterior branch tips, and finally at the top of the crowns [39]. It has caused extensive decline and mortality of eastern hemlock in many parts of the USA, including the southern Appalachian Mountains, portions of the mid-Atlantic region, and southern New England. Because adelgid resistance is rare in hemlock (but see [40,41]), the continued spread of the adelgid threatens to cause the range-wide decline or elimination of this ecologically, culturally, and economically important tree species [42,43].

### 2.2. Effects of the Adelgid on Vegetation Structure and Composition

The adelgid can feed on and kill all sizes and ages classes of eastern hemlock, from the smallest seedling to the largest old-growth tree. Because eastern hemlock trees lack the ability to sprout or recover from chronic needle loss, once the adelgid establishes in a forest stand, chronic tree morbidity and mortality may last for 1–2 decades before it is functionally or completely eliminated [44,45,46,47]. Observations from permanent plots at over 200 locations in central Connecticut and Massachusetts revisited multiple times over the last 20 years have documented the structural and compositional changes that occur in hemlock forests infested with the adelgid [39,48]. The trajectory of canopy thinning and subsequent hemlock mortality has varied by location, but levels of mortality have ranged from 0 to 99% throughout the region.

A complete change in forest cover occurs rapidly following the loss of eastern hemlock throughout New England. Cool and dark conifer-dominated stands with abundant hemlock regeneration shift within a decade to deciduous stands of mixed hardwoods dominated by black birch (*Betula lenta* L; Fagales: Betulaceae), red maple (*Acer rubrum* L.; Sapindales: Sapindaceae), and various oak (*Quercus*; Fagales: Fagaceae) species, along with occasional white pine (*Pinus strobus* L.; Pinales: Pinaceae); new hemlock regeneration is observed only rarely [44,49,50]. Canopy gaps resulting from hemlock mortality also may be colonized by nonnative species such as tree-of-heaven (*Ailanthus altissima* (Mill.) Swingle; Sapindales: Simaroubaceae), oriental bittersweet (*Celastrus orbiculatus* Thunb.; Celastrales: Celastraceae), Japanese stilt grass (*Microstegium vimineum* (Trin.) A. Camus; Poales: Poaceae), and Japanese barberry (*Berberris thunbergii* DC.; Ranunculales: Berberidaceae) [44,49,51]. Bryophytes, often strongly associated with hemlock understory environments, have experienced both increases due to additions of downed wood [52] and strong indirect negative effects of the adelgid through diminished commensal interactions with hemlock and increased amensalistic effects from the deciduous tree species that commonly replace it [53]. 

Pre-emptive salvage logging, a human response to (and hence “indirect impact” of) the adelgid [54], sometimes leads to even larger effects on vegetation. Post-harvesting sites are dominated by *Betula lenta* and *Acer rubrum*, but other more shade-intolerant species, including raspberry (*Rubus* spp.), sedges (*Carex* spp.) and pilewort (*Erechtites hieracifolia* (L.) Raf. ex DC; Asterales: Asteraceae), can precede establishment of woody species [55,56,57,58].

### 2.3. Effects of the Adelgid and Hemlock Decline on Associated Fauna

The deep crowns and understory conditions below hemlock provide important habitat for a variety of animal species, including more than 120 vertebrate species and at least 300 species of insects and other arthropods [3,4,5,6,7,8,9,10,11,12]. Many different bird species, notably warblers (Passeriformes: Parulidae), spend at least part of their life cycle in hemlock forests, often feeding on insects and mites dispersed throughout the dense tree crowns. Species that have exhibited significant declines in mid-Atlantic and New England forests where eastern hemlock is declining or has been lost include the Black-throated green warbler (*Dendroica virens* (Gmelin)), Blackburnian warbler (*Setophaga fusca* (Müller)), and Canadian warbler (*Cardellina canadensis* (L.)), along with the Acadian flycatcher (*Empidonax virescens* (Vieillot); Passeriformes: Tyrannidae) and the Hermit thrush (*Catharus guttatus* (Pallas); Passeriformes: Turdidae) [3,59,60,61]. Another animal strongly associated with hemlock and predicted to be negatively impacted by the loss of hemlock is the white-tailed deer (*Odocoileus virginianus* (Zimmermann); Artiodactyla: Cervidae), which congregates under these evergreens in winter for food and cover [62]. In contrast, moose and deer browse were significantly higher on stems of the abundant tree saplings that are released from competitive suppression by hemlock canopies, or sprout from stumps of hardwoods cut following harvest of hemlock-dominated stands or simulated adelgid decline [63]. Continual herbivory by these ungulates may alter the future trajectory of the succeeding forests’ overstories [64]. There have been fewer studies of small mammal populations in eastern hemlock stands, but Degrassi [65] suggested that decline of eastern hemlock caused by the adelgid is unlikely to affect dramatically small mammal populations. Red-backed salamanders (*Plethodon cenereus* (Green); Caudata: Plethodontidae) and red-spotted newts (*Notophthalmus viridescens* (Rafinesque); Urodela: Salamandridae) thrive under fallen wood in these forests, and the loss of hemlock caused by the adelgid or pre-emptive salvage logging likely will reduce their abundance in the future [66].

Canopy and sub-canopy arthropod community diversity was significantly higher in declining eastern hemlock forests than in intact stands in Connecticut, although ground-level arthropod communities did not differ [67]. Individual arthropod taxa, however, do vary in their response to eastern hemlock decline. For example, loss of eastern hemlocks in Massachusetts and Connecticut caused either by the adelgid or pre-emptive salvage logging was forecast to increase the abundance of ground-dwelling ant species [4]. This prediction was subsequently confirmed experimentally [9,23]. In contrast, Mallis and Rieske [8] found a much richer community of spiders living in eastern hemlock canopies than in deciduous ones in southeastern USA forests. Similarly, Zukswert et al. [58] found higher mites and collembolan densities in soils beneath healthy eastern hemlocks relative to soils in sites where hemlock had been harvested.

Hemlock loss also can impact aquatic fauna. Because eastern hemlock often grows along streams and in wetlands, it modifies abiotic conditions in these aquatic ecosystems by maintaining cooler water temperatures [2]. These conditions are extremely important for trout and their aquatic invertebrate prey; both rely on the thermal modification and stable base flows provided by hemlock [68,69]. As hemlocks have declined along streams, canopy openness and light levels have increased [11,70], often with an increase in variability of water temperature [70] and expected increases in biomass of stream periphyton [71]. Loss of hemlock should increase inputs of woody biomass (as coarse woody debris) to streams [70,72,73] and alter both allochthonous energy inputs and the community structure of benthic detrital shredders [10,74,75,76,77].

In some locations, the loss of hemlock and its replacement by mixed deciduous stands has been observed to lead to a homogenization of regional floras and faunas and a consequent reduction in landscape- or regional-scale beta diversity [7].

### 2.4. Ecosystem-Level Changes

Thinning canopies and eventual tree death resulting from continuous feeding by the adelgid leads to microenvironmental changes such as increased light, soil surface temperatures, and subsurface soil moisture [13,15,20]. These changes lead to cascading changes in ecosystem processes such as accelerated decomposition [15], enhanced nutrient cycling rates and increased nutrient availability [18,20,22,78], and decreased soil respiration [16]. Enhanced nitrogen cycling also can increase nitrogen inputs to streams [79]. Intensive harvesting of eastern hemlock stands also increased rates of decomposition and nitrogen cycling, and nitrogen availability also increased significantly compared to the unharvested portions of the stand [22,55].

The adelgid itself also affects the flow of nutrients and energy from tree canopies to the soil. The woolly wax covering secreted by adult adelgids is rich in C and N [80] and provides energy for increases in foliar microbial populations of yeasts, bacteria, and fungi which, in turn, lead to enriched nutrient content of throughfall as it passes through infested canopies and is deposited on the soil surface [19,81]. Belowground changes are associated with adelgid-caused hemlock decline. Ectomycorrhizal root colonization and abundance of bacteria in the rhizosphere were both significantly reduced in hemlocks infested with the adelgid relative to uninfested controls [82]. In addition, adelgid feeding resulted in significantly greater root mass fraction [83] and lower shoot:root biomass ratios [84].

Hemlock loss also impacts hydrological cycles. The recent decline of hemlock in New England [85] has increased water yield as the declining trees take up less water [86]. Over the long term, however, observations from the southern Appalachians [46] suggest that this trend may be reversed. If deciduous species with higher summer water use, including *B. lenta* and *Q. rubra*, replace hemlock, small headwater streams may experience reduced yield and dry out during summer [14,87].

## 3. Adelgid Spread Models

Several studies have analyzed and modeled patterns of spread of the adelgid or its effects on hemlock forests across their geographic range in eastern North America.

### 3.1. Understanding and Predicting the Spread of the Adelgid

County-level spread records of the adelgid have been used to investigate associations between forest structure and abiotic factors, and spatial variation in adelgid spread rates [88,89,90]. For example, Morin et al. [88] used non-spatial linear regression to examine the role of hemlock abundance and January minimum temperatures in explaining spatiotemporal variation (anisotropy) in spread rates in different cardinal directions from the initial infestation location. They found that spatial variation in spread rate was positively related to hemlock abundance but negatively related to January minimum temperature, a finding also highlighted by Evans and Gregoire [90]. Fitzpatrick et al. [89] used a spatially explicit Bayesian hierarchical framework and found evidence that patterns of adelgid spread could be explained by human population density, mean winter temperature, and hemlock abundance, except along the southeastern and northernmost edges of the spread boundary, where spread of the adelgid proceeded more slowly than expected. 

Fitzpatrick et al. [33] used insights from these and other studies to develop a spatially explicit, stochastic model of adelgid spread that combined field-based estimates of dispersal and population processes with raster maps characterizing spatiotemporal heterogeneity in climate and hemlock abundance. This model incorporated four factors known to influence the spread of the adelgid: (i) abundance of eastern hemlock; (ii) mean winter (December-January-February-March) temperature; (iii) population growth of the adelgid [91]; and (iv) its observed dispersal dynamics. Hemlock abundance and winter temperature were represented using rasters comprised of 1 × 1-km cells. Hemlock abundance in each cell determined the probability that dispersing adelgids established in that cell and set the upper limit to population growth once the cell became infested. Hemlock abundance declined annually in infested cells to mimic tree mortality and remained constant throughout the simulation in uninfested cells. Mean winter temperatures varied annually following observed temperature fluctuations during the simulation period and influenced population growth by limiting the proportion of overwintering adelgids that survived to produce offspring in the next spring. To simulate population dynamics in each hemlock stand, multiyear surveys of adelgid reproduction and survival rates in Massachusetts and Connecticut were used to estimate mean parameter values for each of its life stages. For the progrediens generation, these parameters included average number of progrediens produced by each overwintering sistens and the mortality rate of the progrediens. For the sistens generation, parameters included the average number of sistens produced by progrediens and the mortality rates of dispersing, aestivating, and overwintering sistens. Dispersal between cells was simulated using two processes: (i) local diffusion between neighboring cells; and (ii) long-distance dispersal based on a function parameterized from multiple datasets documenting the spread of the adelgid across different regions of the eastern USA (Table 2 in [33]). See [33] for more details regarding the spread simulation and access to the code for reproducing the simulations.

The spread simulation described in [33] began in 1951 with an initial infestation near Richmond, Virginia, where the adelgid was first documented in eastern North America [92] and proceeded for 58 annual time steps (through 2008). The model was run for 1000 stochastic simulations to obtain an average representation of adelgid spread during 1951–2008. To evaluate the model, the observed pattern of adelgid spread from the USFS county-level dataset [93] was compared to model predictions. The results of these 1000 stochastic simulations were used as the starting conditions for the forecasts described below (Section 4.1).

The model simulated spatial and temporal heterogeneity in the probability and timing of adelgid spread across the geographic range of hemlock and suggested that winter temperature was more important than hemlock abundance in determining anisotropic spread rates. Under the contemporary winter temperature regimes that formed the basis of the simulation, the model predicted that the northern third of the range of hemlock was at very low or no risk of invasion, with the northern and southern predicted limits of spread generally matching the observed extent in 2008. The model did not, however, correctly predict the timing of the first reported adelgid infestation in most counties, which Fitzpatrick et al. [33] attributed to the challenges of modeling rare, long-distance dispersal events early in the invasion that substantially affected the ensuing pattern of spread.

### 3.2. Forecasting the Ecosystem Consequences of Hemlock Loss

Albani et al. [94] combined a stochastic model of the adelgid spread (parameterized with occurrence data collected through 2004) with the ecosystem-demography (ED) model of forest ecosystem dynamics [95] to forecast the spread and impacts of the adelgid on forest composition and carbon exchange. Their stochastic spread model forecast that the adelgid would arrive in northern Minnesota by 2030 [94]. Their forest-dynamics model forecast hemlock loss caused by the adelgid initially would reduce forest carbon uptake by 8%, with the most severe impacts occurring in areas with high hemlock density (western Pennsylvania, portions of New York, Massachusetts, Vermont, New Hampshire, and southern Maine; Figure 1). Albani et al. [94] concluded that the impact of the adelgid on the carbon (C) cycle could be substantial locally but small regionally. Models of the effects of the adelgid on net primary productivity and heterotrophic respiration in dense hemlock stands at Harvard Forest and Hubbard Brook (New Hampshire) did not forecast that the adelgid could shift these stands from C sinks to sources [94,96]. In fact, we already have observed that declining hemlock stands at Harvard Forest transitioned from being C sinks to C sources in less than a decade after the adelgid was first detected there [97] (Figure 2).

As early successional hardwood species replace hemlock, C uptake has been forecast to increase rapidly and eventually surpass that of the intact hemlock stands [94,96,98,99,100]. These changes in ecosystem dynamics will take decades as hemlock forests transition slowly to forests dominated by mixed hardwood species.

Associated changes in foliar chemistry, microenvironment, and decomposition in emerging deciduous forests are expected to lead to reduced soil organic matter content and smaller total soil C pools and increases in aboveground net primary productivity and soil nitrogen uptake rates [96,98,99,100,101]. These predictions have been incorporated into several forecasts of the effects of adelgid-caused hemlock loss on total standing C stocks in declining hemlock stands and the early successional hardwoods that replace them. Raymer et al. [98] used data collected through 2010 on the Prospect Hill tract at Harvard Forest [102], in the Harvard Forest Hemlock Removal Experiment [103], and across Connecticut [44] to forecast trends in C storage in changing New England forests. They found that these forests would be resilient to hemlock loss in the short term because the increase in coarse woody debris (CWD) from dead and dying hemlocks plus growth by recruiting and released black birch saplings and young trees would approximately equal the decrease in C in live hemlock trees.

Nonetheless, Raymer et al. [98] forecast an overall net loss of ecosystem C content by 2100 because it takes ≈20 years for C stocks in black birch stands to reach pre-adelgid levels and there is an ≈40-year difference between maturing secondary hemlock and black birch stands in their accumulation of C stocks compared to older (“primary”) hemlock stands. The ≈20% difference in total C stocks after 100 simulated years between intact hemlock stands and those that had been killed by the adelgid and replaced by successional hardwoods can be conceptualized as the loss of compound interest. Because the adelgid removes the “principal” (existing hemlock stands) and it takes 20–40 years of regrowth to reach the same standing crop as was removed, the potential growth of hemlock in that 20–40-year interval (“interest”) is foregone [98]. This forecast was robust, with outcomes varying little as a function of which mid-successional hardwood species replaces hemlock—black birch in southern New England [98], red maple in northern New York and New England [100], or yellow birch (*Betula alleghaniensis* Britt.) in New York’s Catskill Mountains [99]—or the model used—allometric equations [98], ED [94], Forest Vegetation Simulator [100], or the new Spe-CN [99].

## 4. New Forecasts

The predictions of the spread of the adelgid [33] served as the jumping off point for our future forecasts. Those simulations ended in 2008, with regions of southern New York and New England predicted to have very low probabilities (<1%) of infestation (even though these regions had, in fact, been infested by the adelgid for many years). Northern New England, Wisconsin, the upper peninsula of Michigan, and southern Canada were predicted to have zero probability of infestation (largely consistent with the observed pattern of spread at that time). Application of the model starting in 2008 provided us with a new opportunity to compare predictions with the observed pattern of spread by 2016 and to update our inferences about how the spread of the adelgid and hemlock decline may play out through 2050.

### 4.1. Will the Adelgid Continue to Spread?

To forecast potential future spread of the adelgid and associated hemlock loss, we used outputs from the simulations of Fitzpatrick et al. [33] and started a new set of simulations than ran from 2009 to 2050. These new simulations took as input (1) the final (predicted) locations and population sizes of the adelgid in 2008 averaged across the 1000 contemporary simulations reported in [33]; and (2) raster maps of eastern hemlock abundance reflecting its mortality resulting from adelgid infestations, also averaged across the 1000 contemporary simulations reported in [33]. We then ran new simulations in annual time steps for the period 2009 to 2050. To estimate future winter temperatures, we used annual forecasts of mean winter temperatures from the World Climate Research Program’s (WCRP’s) Coupled Model Intercomparison Project phase 3 (CMIP3) multi-model dataset [104]. To describe a range of future conditions, we selected three combinations of global circulation models and emission scenarios that produced high, middle, and low warming forecasts, including MICRO3.2 forced using the A2 scenario (high warming), CNRM-CM3 forced using the A1b scenario (midrange warming) and CCSM3 forced using the B1 scenario (low warming). Initial tests confirmed that simulations of future spread were much less variable for future projections than were simulations started in 1951 by [33]. This was to be expected, because the 1951–2008 simulations began with a single infested location and therefore subsequent spread dynamics were highly sensitive to stochasticity. In contrast, our 2009–2050 forecasts exhibited very little between-simulation variability (Figure 3) because we used 2008 starting conditions that already reflected the average of 1000 58-year spread trajectories. For this reason, we ran only 100 stochastic simulations per climate forecast.

To evaluate model predictions, we followed the same procedure as described in Fitzpatrick et al. [33], but here only evaluated the simulations in terms of temporal accuracy (since spatial accuracy was largely confirmed in [87]) using an updated county-level adelgid spread map for 2017 [93]. Briefly, for each newly infested county since 2008, we obtained the predicted years of first infestation across all cells within the county and across all 100 spread simulations for each climate forecast. We then determined whether the 95% confidence interval of the distribution of predicted years of infestation contained the year the adelgid was first reported in that county.

By 2015, our new model predicted that nearly all hemlock stands south of the Vermont- Massachusetts border would have a 100% probability of adelgid infestation (Figure 4a). The model predicted a low infestation probability for stands north of this border. These low-probability sites included the Adirondacks, where the adelgid is now established, and southern Vermont and New Hampshire, where it already had been observed in the early 2010s. The model also predicted very little, if any, risk of infestation in the upper Midwest states by 2015 (although the adelgid in fact had reached western Michigan by 2015). By 2030, however, the new model forecasts that nearly all of New England, except for northeastern-most Maine and portions of southern Canada, will be at very high risk for adelgid infestation (Figure 4b). Concurrently, most stands in Michigan and portions of eastern Wisconsin are forecast to have low (but non-zero) infestation risk. By 2050, as projected warming throughout this region largely eliminates winter temperature constraints on adelgid populations, the adelgid is forecast to infest nearly all hemlock stands in its range (probability > 0); notable exceptions include the northernmost stands in Canada, Wisconsin, and Minnesota (Figure 4c).

#### Model Evaluation

Of the 126 counties that became infested by the adelgid after 2008, the observed year of first infestation fell within the 95% confidence interval of the simulated year for 27, 26, and 27 counties for the A2, A1B, and B1 CMIP3 scenarios, respectively (Figure 5). Most counties for which the model correctly predicted the observed year of first infestation (Figure 5, yellow shading) fell along the western and southern edge of the invasion front. As in [33], the model tended to predict later arrival (Figure 5, red shading) in New England and western Michigan than observed, up to a maximum of 25 years later (in Ottawa County, Michigan under the B1 scenario). Earlier-than-observed arrivals (Figure 5, red shading) were aligned mainly along the western front of the invasion, up to a maximum of eight years earlier in several counties in Pennsylvania and Tennessee. 

### 4.2. Projected Losses of Hemlock from Eastern Forests

Adelgid densities fluctuate in response to both winter temperatures and density-dependent changes in the nutritional quality of hemlock [105]. As tree health and availability of new branch and needle growth declines when the adelgid feeds on it, the population density of the adelgid declines in parallel. Nutritional quality of hemlocks continues to decline from when a tree becomes infested until it is defoliated and dies. Our simulations captured the process of hemlock decline and feedback with adelgid populations using concepts from radioactive decay. Stands that remained heavily infested had a shorter half-life and declined faster than stands where adelgid population growth was limited by winter temperatures; the parameters for this relationship that we used in our simulations corresponded with observed patterns of hemlock decline [39]. Note that our simulations did not account for other potential controls on adelgid population dynamics, such as other aspects of climate or biological control.

Once a hemlock stand became infested in our simulations, hemlock abundance declined in that stand as a function of adelgid population density; greater adelgid densities led to more rapid loss of hemlock. Adelgid density was, in turn, determined by annual fluctuations in winter temperature and the amount of available hemlock, such that hemlock stands in regions where winter temperatures limit adelgid populations may be infested for long periods of time but not suffer large hemlock declines while winter temperatures limit adelgid populations. This effect can be seen in the simulation results. By 2015, even though most hemlock stands south of Vermont were predicted to have been infested, the model simulated very little loss of hemlock basal area in this region (Figure 4d). Hemlock was also predicted to remain throughout its southern range up to 2015, but at greatly reduced basal area. By 2030, the model forecasts that hemlock will be almost entirely lost from forests south of southern New England (Figure 4e). By 2050, healthy hemlock stands are forecast to be limited to northernmost New England, high elevation stands in the Adirondacks, the upper peninsula of Michigan, and Wisconsin, and stands along hemlock’s northern range limit in Canada (Figure 4f). Hemlock is forecast to be almost completely lost from forests south of these areas by 2050.

### 4.3. Projected Changes in Ecosystem Dynamics in Eastern Forests

No new forecasts of ecosystem dynamics (e.g., C or N pools or fluxes) in declining hemlock forests that take advantage of data collected after 2010 have been published. Unlike previous models [94,98,99,100], any new models would need to account for the real possibility that declining hemlock stands can shift rapidly from being a C source to a C sink (Figure 2).

### 4.4. Interactions between the Adelgid and Other Species

Interactions between the adelgid and other species also may play an important role in determining its distribution, spread, and effects on hemlock [106]. Such interspecific interactions appear particularly likely between the adelgid and the elongate hemlock scale (*Fiorinia externa* Ferris; Hemiptera: Diaspididae), another nonnative species that feeds primarily on eastern hemlock [107]. *Fiorinia* was first detected in Long Island NY in 1908 [108], and in the 1990s it began to expand its range rapidly into adelgid-infested hemlock stands in southern New England [38,106]. The elongate hemlock scale competes exploitatively with the adelgid for plant photosynthate and reduces adelgid densities by 30–40% when the two species co-occur [109,110]. Both field and laboratory work also has shown that mobile adelgid crawlers avoid settling at the base of hemlock needles already colonized by the scale [111]. Recent work comparing models of range expansion of the adelgid and the scale found that prior adelgid presence was linked to higher scale densities [112]. Although there was no corresponding impact of scale on adelgid distribution, high densities of the scale may eventually affect landscape-level patterns of adelgid colonization and population growth.

## 5. Discussion

In much of its range south of New England, the hemlock woolly adelgid has led to declines in hemlock [43], and in some cases removed more than 90% of the overstory density in forests it dominated through the 1980s [44,45,46,47]. In New York, New England, the upper Midwest, and southern Canada, the adelgid likely will do the same to hemlock there in the next 20 years. Decades of research on the consequences of hemlock loss have consistently revealed the myriad ways in which the loss of this foundation species (sensu [2]) will alter local- and landscape-level forest structure, biological diversity, and ecosystem dynamics [42]. Modeling efforts using different assumptions and implementations have tended to underestimate consistently and in similar ways the rate of adelgid spread [33,88,89,94] and are just beginning to incorporate its interactions with other insect species [113]. Testing these models with long time series of data on the spread of the adelgid has led to their revision and refinement (Figure 4), although we still tend to under-predict its spread to the north and east (Figure 5). It remains unclear whether these underpredictions result from inadequately modeling accelerating climate change or failing to account for people inadvertently moving the adelgid into new locales. Further examination of the role of transportation corridors and movement of plants by the horticultural trade should be included in further cycles of modeling, testing, and model revision as the adelgid expands into the northern- and westernmost range of *T. canadensis*.

Similarly, models of ecosystem-wide consequences of hemlock decline executed on different modeling platforms and parameterized with different starting conditions have yielded consistent results. In the absence of the adelgid or other external agents of forest decline, the net primary productivity and C storage of northeastern US forests is forecast to increase as the climate warms [113]. However, this same region is the Earth’s “ground zero” for forest pests and pathogens [114]. Models of ecosystem impacts of the adelgid (and other outbreaking insects and pathogens) have suggested that their impacts on standing C stocks may be minimal (≈20% in foregone accumulated “interest”; [99]), but these models have all assumed that the successional transition from hemlock (or other late successional species) to early successional hardwoods will be slow and steady. At Harvard Forest, however, hemlock decline is occurring faster than expected, and our oldest stand of eastern hemlocks has, contrary to model predictions [94,96,98], shifted from being a C source to a C sink (Figure 2). New generations of ecosystem models, or at least reparameterizing of existing models with current and emerging time series of data, are needed to improve forecasts of the effects of the adelgid on the dynamics of the future forests of eastern North America.

## 6. Conclusions

The hemlock woolly adelgid, first detected in eastern North America in 1951, now occurs throughout a substantial portion of the range of eastern and Carolina hemlock. The adelgid has spread anisotropically and more rapidly than had been forecast using either stochastic or climatically-driven spread models. The role of unique initial conditions in first-generation dispersal models has been ameliorated in new models that use more recent data and ensemble averages of previous model runs. Although a new model does a better job of predicting the spread of the adelgid in the south and west of the range of hemlock, it still under-predicts its spread in the north and east. The effects of the adelgid on hemlock forest ecosystems has been pronounced. The adelgid has reset successional sequences, homogenized biological diversity at landscape scales, altered hydrological dynamics, and shifted forest stands that once were carbon sinks into carbon sources. Ecosystem models of adelgid-driven hemlock dynamics have consistently forecast that carbon stocks will be affected little by the shift from late-successional dominance by a foundation species to early successional mixed hardwood stands, but these forecasts have assumed that the successional transition will be slow and steady and that the intermediate stages will remain carbon sinks. New forecasting models of adelgid-driven hemlock decline need to account for observed abrupt changes in carbon flux, ongoing human-driven land-use changes, and accelerating climatic change.

## Figures and Tables

**Figure 1 insects-09-00172-f001:**
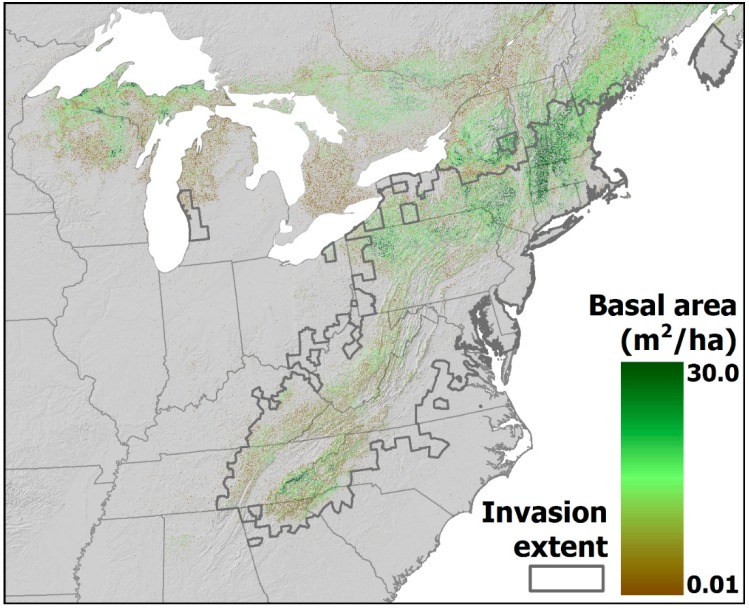
Range and abundance (as basal area in m^2^/ha) of eastern hemlock (*Tsuga canadensis*) in eastern North America (data from [33]). The extent as of 2017 of the nonnative hemlock woolly adelgid *(Adelges tsugae*) is indicated by the thick grey line.

**Figure 2 insects-09-00172-f002:**
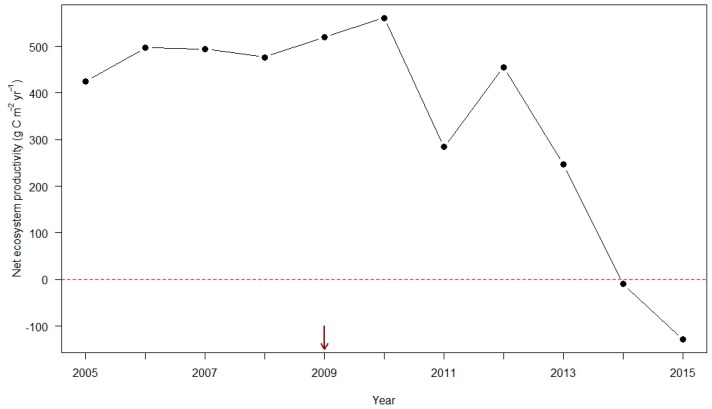
Net ecosystem productivity (NEP) estimated by Marc-André Giasson from measurements taken at an eddy-covariance tower located in a >200-year-old hemlock stand at Harvard Forest, Petersham, Massachusetts [97]. NEP > 0 (red-dashed line) indicates that the forest stand is a net carbon sink, whereas NEP < 0 indicates that the forest stand is a source of carbon to the atmosphere. The dark red arrow indicates the year in which the adelgid infestation was first convincingly observed in this stand.

**Figure 3 insects-09-00172-f003:**
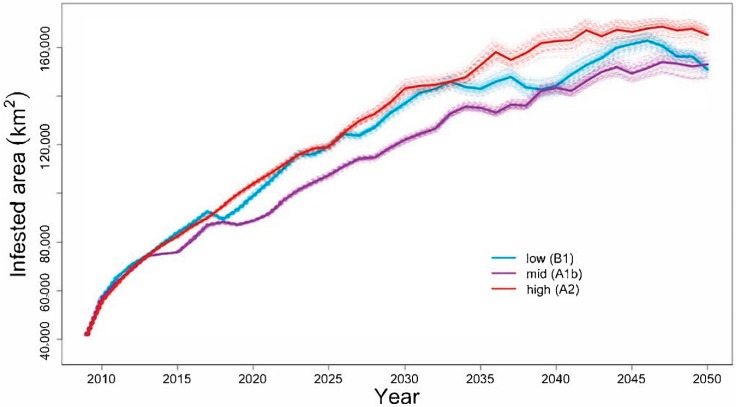
Forecast area (in km^2^) of infestation of the hemlock woolly adelgid through time for three different climate-change scenarios. The solid line represents the average of each of 100 individual simulations (light, dashed lines).

**Figure 4 insects-09-00172-f004:**
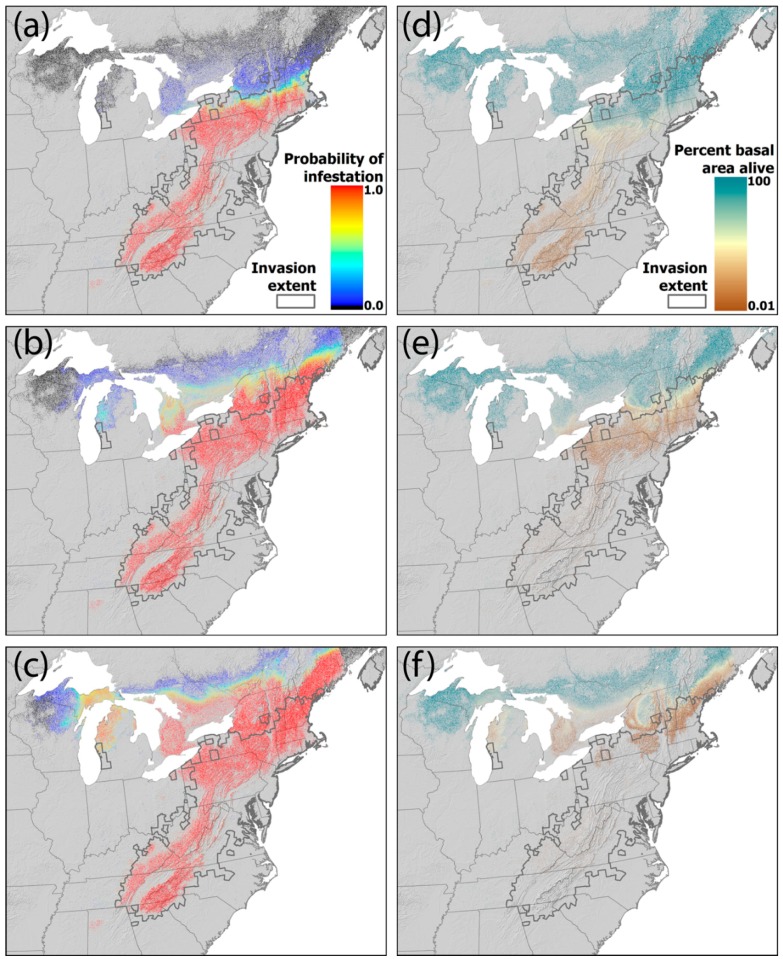
(**a**–**c**) Predicted probability of infestation of hemlock stands by the hemlock woolly adelgid and (**d**–**f**) estimated resulting loss of hemlock basal area for (**a**,**d**) 2015, (**b**,**e**) 2030, and (**c**,**f**) 2050. Maps show results for the three climate-change scenarios (A2, A1b, B1) averaged over 100 simulations. The observed extent of the infestation in 2017 is delineated by the bold gray line. Light gray regions do not contain hemlock. Note that the abrupt line visible in the upper right corner of each panel reflects the extent of future climate simulations, which did not extend much beyond northeastern Maine.

**Figure 5 insects-09-00172-f005:**
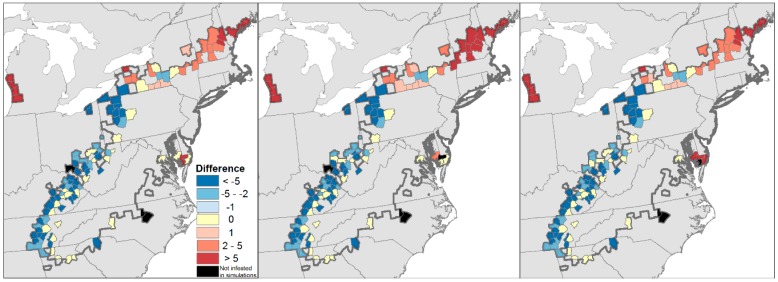
Maps of counties infested by the hemlock woolly adelgid between 2009 and 2017 and the correspondence between the predicted and observed year of first infestation for the (left) A2, (center) A1b, and (right) B1 future climate-change scenarios. Yellow shading indicates counties for which the observed and simulated year of first infestation did not differ (i.e., the observed year fell within the 95% confidence interval of the simulated year). Blue or red shading indicates counties for which the model predicted a county to become infested earlier or later, respectively, than was observed. Black shading represents counties where the adelgid has been observed, but for which none of the 100 simulations predicted infestation. The observed extent of the invasion in 2017 is delineated by the bold gray line. Note that counties in Canada infested by hemlock woolly adelgid are not shown, as they fall outside the domain of future climate scenarios.

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
