# Peer review of "The Past, Present, and Future of the Hemlock Woolly Adelgid (Adelges tsugae) and Its Ecological Interactions with Eastern Hemlock (Tsuga canadensis) Forests"

_insects, 2018, doi:10.3390/insects9040172_

Round 1
Reviewer 1 Report
A useful exercise in understanding the impacts and spread of HWA. An updated prediction has value on a number of different levels. I appreciate the effort made here.
I do have some comments:
Lines 64-65. This is factually incorrect statement that continues to get perpetuated. It was first discovered in the early 1950’s. No one knows exactly when it was introduced. Please re-read the literature. Reference 34 does not state when the insect was introduced or where!!! Syoetzel in a 2002 proceedings abstract states it was firest collected in Richmond, VA in 1951. It is believed to have come in on horticultural plantings in the early 20th century when Japanese gardens were popular. Please read Havill et al. 2016, as it best describes our knowledge on its introduction.
(Havill NP, Vieira LC, Salom SM (2016) Biology and control of hemlock woolly adelgid. FHTET-2014-05. United States Department of Agriculture, United States Forest Service Morgantown, West Virginia)
Line 409, Change this statement about introduced in 1951 to found in 1951.
General comments.
L. 192-212. Citing and describing Fitzgerald et al 2012 and then revising it with updated data later, seems inadequate to this reader. Since so much of the prediction work is based on this model, it should be presented, citing the original article. The authors may argue, that I should be able to go back and find Fitzgerald et al 2012 and look up the formula……. Which I did and I see it is coded in R in the appendix. I’m not sure the resolution, but I’m not big on relying on faith in terms of making new predictions from a model that is embedded in an appendix of another article.
L. 378-379. I’m not sure this statement can be made quite yet. Ongoing sampling of hemlock forests show that they have either disappeared or are declining steadily. No question that some stands are functionally irrelavent, but most to all is a bit of an overstatement. Can this statement be softened just a bit? It is not like any data were presented to support it.
L. 410. Looking at Fig 3a, I’m not sure how the authors come up with HWA being present in 90% of the range of hemlock.
Figure 4 is somewhat confusing. Can the authors clarify the figure caption? What is observed and what is predicted? Red observed and black predicted?? Or observed is the gray line and red is within and black is outside. Not intuitive to me.
Author Response
A useful exercise in understanding the impacts and spread of HWA. An updated prediction has value on a number of different levels. I appreciate the effort made here.
RESPONSE: Thank you very much.
I do have some comments:
RESPONSE: We thank the reviewer for their useful comments and have revised the manuscript accordingly.
Lines 64-65. This is factually incorrect statement that continues to get perpetuated. It was first discovered in the early 1950’s. No one knows exactly when it was introduced. Please re-read the literature. Reference 34 does not state when the insect was introduced or where!!! Syoetzel in a 2002 proceedings abstract states it was firest collected in Richmond, VA in 1951. It is believed to have come in on horticultural plantings in the early 20th century when Japanese gardens were popular. Please read Havill et al. 2016, as it best describes our knowledge on its introduction.
(Havill NP, Vieira LC, Salom SM (2016) Biology and control of hemlock woolly adelgid. FHTET-2014-05. United States Department of Agriculture, United States Forest Service Morgantown, West Virginia)
RESPONSE: We have changed the language from “introduced” to “detected” and added the citation.
Line 409, Change this statement about introduced in 1951 to found in 1951.
RESPONSE: We have changed the language from “introduced” to “detected”.
General comments.
L. 192-212. Citing and describing Fitzgerald et al 2012 and then revising it with updated data later, seems inadequate to this reader. Since so much of the prediction work is based on this model, it should be presented, citing the original article. The authors may argue, that I should be able to go back and find Fitzgerald et al 2012 and look up the formula……. Which I did and I see it is coded in R in the appendix. I’m not sure the resolution, but I’m not big on relying on faith in terms of making new predictions from a model that is embedded in an appendix of another article.
RESPONSE: Sorry for any confusion. We added some more details regarding population and dispersal dynamics in the model but completely describing an already published model would significantly increase the length of the article. We feel adding such detail in the new manuscript is not warranted as Fitzpatrick et al. (2012) is a peer-reviewed article that fully details the spread model. The model used in this study was not revised in any way from the original publication, we simply restarted the simulation using the ending conditions from Fitzpatrick et al. (2012) and applied it to a new set of winter temperature data extracted in annual time steps from global circulation models to forecast the pattern of spread expected under warmer winters. Thus, in this manuscript, we provide only the most salient details needed to understand the major components of the spread model and its predictions, and we assume that readers interested in more details can read the original publication or download the code and re-run it.
L. 378-379. I’m not sure this statement can be made quite yet. Ongoing sampling of hemlock forests show that they have either disappeared or are declining steadily. No question that some stands are functionally irrelavent, but most to all is a bit of an overstatement. Can this statement be softened just a bit? It is not like any data were presented to support it.
RESPONSE: We have softened the language as requested.
L. 410. Looking at Fig 3a, I’m not sure how the authors come up with HWA being present in 90% of the range of hemlock.
RESPONSE: We have edited the sentence as follows: “The hemlock woolly adelgid, first detected in eastern North America in 1951, now occurs throughout a substantial portion of the range of eastern and Carolina hemlock.”
Figure 4 is somewhat confusing. Can the authors clarify the figure caption? What is observed and what is predicted? Red observed and black predicted?? Or observed is the gray line and red is within and black is outside. Not intuitive to me.
RESPONSE: We have provided a new figure (note that it is now Figure 5) and edited the caption to clarify the presentation.

Reviewer 2 Report
Review comments for manuscript Insects-384878
The manuscript titled “The past, present, and future of the hemlock woolly adelgid (Adelges tsugae) and its ecological interactions with eastern hemlock (Tsuga canadensis) forests.
General Summary and Comments: The manuscript is generally organized into three sections. In the first (section 2 in the text), the authors focus on reviewing the status and history of the hemlock woolly adelgid in eastern North America, discussing both the patterns which have been observed over the period (generally) from 1950 to 2008. The second portion of the manuscript (section 3) focuses on reviewing and summarizing some of the modeling work that has been carried out with regards to both the spread of the adelgid, and its impact on ecosystem processes (primarily carbon dynamics). In the third section (section 4 in the text) a new model is presented with an emphasis on forecasting adelgid impacts and movement on the landscape.
Overall, this information is likely to be of interest to a variety of audiences including natural resource managers, invasive species biologists, and biogeographers. The first section (section 2 in the text) provides a well-written review of much of the research that has been done on the adelgid, and this summary is likely to be of interest. The second part (section 3 in the text) also does a good job of summarizing some of the modeling work that has been carried out. Both of these sections are well structured, though both have similar problems regarding the language used in the text (refer to section below regarding the use of potentially overstating some issues). The third section, regarding the development of a “new” model may be of value, but it is difficult to determine its value based on the information the authors have provided. More detail regarding this issue is provided below. The manuscript also includes a number of awkward phrases and text, though these are minor edits. With these issues addressed, the paper would seem well suited to Insects.
Potentially overstating some cases:
The text provides a good summary of much of the work that has been done on the hemlock woolly adelgid, and provides an updated review for the topic, something which is of definite value. However, the authors include some text that seems out of place in a peer-reviewed document, specifically, the authors make a number of bold statements which do not seem to be supported by the literature. The following are examples:
Abstract (line 14): The first line of the abstract states “The nonnative hemlock woolly adelgid is steadily eliminating eastern hemlock trees from eastern North America.” Many researchers and land managers would likely agree that the adelgid poses a very serious threat to eastern forests. However, this first sentence seems to imply that the hemlocks of eastern North America are moving consistently towards extinction. This is probably overstating the case. It is worth noting that previous authors argued as early as the late 1980s that the hemlock would be lost quickly, though they still remain. Having a good hook sentence is important, but overstating the issue can create new issues. Arguably, the abstract does not need this first sentence.
Line 50: “…poised to disappear,…” again, this sounds like hyperbole, and the text does not provide citations to support it. Consider removing or changing to less extreme language.
Lines 378 – 379: The text states “In much of its range south of New England, the hemlock woolly adelgid has rendered hemlocks for the most part, functionally irrelevant in forests it dominated through the 1980s.” Similar language is used in lines 79-82. No citation is provided for these statements, and these are arguably overstated. Hemlock continues to be well distributed throughout its southern range, and is still a stand dominant in many locations. It is threatened, and this is a point worth making, but not overstating.
Lines 150-152: The authors state “In all cases, the loss of hemlock and its replacement by mixed deciduous stands has been observed to lead to … … and a consequent reduction in landscape- or regional-scale beta diversity [7].” This issue remains unknown, and the cited paper includes the statement “This research is exploratory, however, and tests of these predictions across larger spatial scales will be necessary to determine the generality of the findings.”
The use of a new model:
The authors cite the model developed in Fitzpatrick et al. 2012, which analyzed patterns from 1951 through 2008, and extend this model by adding two new time periods. The first is the time from 2008 through 2016, which is now historical data. This data is essentially used as a validation data set, though the authors do not seem to describe it as such. The second time period is the period from 2016 through 2050, when the model is used as a forecasting tool. It could be argued that this is not a new model, but simply the use of an existing model as a forecasting tool (using three climate scenarios), however it is important to note that while this may not be a new model, it is still an important analysis. The description of the modeling exercise could benefit from some additional information including:
Why were only 100 iterations, even if stochasticity is reduced? Computer time is pretty cheap these days, using 1000 iterations simplifies making the comparisons with the previously published data.
Section 4.2 discusses links between adelgid density and hemlock loss, but the statements are qualitative and do not seem to include quantitative components for the link, or citations for what the links are. How were these parameters defined? Is a validation data set used for this part of the model? These are important issues, and could be included as a supplemental section. Without this information, this section is difficult to assess.
Miscellaneous and minor issues:
The legend for Figure 3 states “…for (a, c) 2015,…” should this be (a, d)?
Line 386 refers to “Confronting these models…”, the term “validating” might be suitable.
In lines 64-65, and on lines 409-410, the text indicates the adelgid was introduced to eastern North America, however, this is the time when it was first documented in eastern North America. The date of its introduction is not known.
Line 391, why is wood movement listed? While this is a path for the movement of wood boring insects (bark beetles, Emerald Ash Borer, Asian Longhorned Beetle, etc), is it a pathway for the adelgid?
Line 394 – 396, consider providing a table of the models to summarize their results.
Line 263, should the text “…C stocks 100 years…” be “…C stocks after 100 years…”?
Line 44, the text indicates hemlock has been dominant, though I am not sure that is the case, perhaps “significant”, or “common”?
Line 142, consider replacing “of” with “in”
Lines 73 – 75, consider adding citation for Morin and Liebhold 2015, Forest Ecology and Management 341:67-74.
Author Response
General Summary and Comments: The manuscript is generally organized into three sections. In the first (section 2 in the text), the authors focus on reviewing the status and history of the hemlock woolly adelgid in eastern North America, discussing both the patterns which have been observed over the period (generally) from 1950 to 2008. The second portion of the manuscript (section 3) focuses on reviewing and summarizing some of the modeling work that has been carried out with regards to both the spread of the adelgid, and its impact on ecosystem processes (primarily carbon dynamics). In the third section (section 4 in the text) a new model is presented with an emphasis on forecasting adelgid impacts and movement on the landscape.
Overall, this information is likely to be of interest to a variety of audiences including natural resource managers, invasive species biologists, and biogeographers. The first section (section 2 in the text) provides a well-written review of much of the research that has been done on the adelgid, and this summary is likely to be of interest. The second part (section 3 in the text) also does a good job of summarizing some of the modeling work that has been carried out. Both of these sections are well structured, though both have similar problems regarding the language used in the text (refer to section below regarding the use of potentially overstating some issues). The third section, regarding the development of a “new” model may be of value, but it is difficult to determine its value based on the information the authors have provided. More detail regarding this issue is provided below. The manuscript also includes a number of awkward phrases and text, though these are minor edits. With these issues addressed, the paper would seem well suited to Insects.
We thank the reviewer for their useful comments and have revised the manuscript accordingly.
Potentially overstating some cases:
The text provides a good summary of much of the work that has been done on the hemlock woolly adelgid, and provides an updated review for the topic, something which is of definite value. However, the authors include some text that seems out of place in a peer-reviewed document, specifically, the authors make a number of bold statements which do not seem to be supported by the literature. The following are examples:
Abstract (line 14): The first line of the abstract states “The nonnative hemlock woolly adelgid is steadily eliminating eastern hemlock trees from eastern North America.” Many researchers and land managers would likely agree that the adelgid poses a very serious threat to eastern forests. However, this first sentence seems to imply that the hemlocks of eastern North America are moving consistently towards extinction. This is probably overstating the case. It is worth noting that previous authors argued as early as the late 1980s that the hemlock would be lost quickly, though they still remain. Having a good hook sentence is important, but overstating the issue can create new issues. Arguably, the abstract does not need this first sentence.
RESPONSE: For a non-specialist, it is useful to know why one might study the adelgid at all. So the first sentence sets that up. However, we have softened the language by changing “eliminating” to “killing” and “from eastern North America” to “in many parts of eastern North America”.
Line 50: “…poised to disappear,…” again, this sounds like hyperbole, and the text does not provide citations to support it. Consider removing or changing to less extreme language.
RESPONSE: We have softened the language here.
Lines 378 – 379: The text states “In much of its range south of New England, the hemlock woolly adelgid has rendered hemlocks for the most part, functionally irrelevant in forests it dominated through the 1980s.” Similar language is used in lines 79-82. No citation is provided for these statements, and these are arguably overstated. Hemlock continues to be well distributed throughout its southern range, and is still a stand dominant in many locations. It is threatened, and this is a point worth making, but not overstating.
RESPONSE: We have revised the first paragraph of the Discussion in its entirety, and revised language in the Introduction, too. Relevant citations in both locations are [44 – 47].
Lines 150-152: The authors state “In all cases, the loss of hemlock and its replacement by mixed deciduous stands has been observed to lead to … … and a consequent reduction in landscape- or regional-scale beta diversity [7].” This issue remains unknown, and the cited paper includes the statement “This research is exploratory, however, and tests of these predictions across larger spatial scales will be necessary to determine the generality of the findings.”
RESPONSE: We have changed the opening of the sentence from “In all cases” to “In some locations”.
The use of a new model:
The authors cite the model developed in Fitzpatrick et al. 2012, which analyzed patterns from 1951 through 2008, and extend this model by adding two new time periods. The first is the time from 2008 through 2016, which is now historical data. This data is essentially used as a validation data set, though the authors do not seem to describe it as such. The second time period is the period from 2016 through 2050, when the model is used as a forecasting tool.
RESPONSE: To be clear, we only ran the model for one time period 2009-2050. We did not use observed climate for the 2009-2017 period for which we have validation data because we wanted to see how well the model might have predicted spread under assumptions of future climate.
It could be argued that this is not a new model, but simply the use of an existing model as a forecasting tool (using three climate scenarios), however it is important to note that while this may not be a new model, it is still an important analysis.
RESPONSE: We agree 100%. The model has not been modified in any way, we simply use it in a forecasting context, whereas Fitzpatrick et al (2012) used it for prediction.
The description of the modeling exercise could benefit from some additional information including:
Why were only 100 iterations, even if stochasticity is reduced? Computer time is pretty cheap these days, using 1000 iterations simplifies making the comparisons with the previously published data.
RESPONSE: We feel there are few additional insights that would be gained from running the model for 1000 stochastic simulations. The variability between simulations for the future was miniscule (see new Figure 3 in ms.). While we agree that computer time is relatively cheap, working with and analyzing output from 3000 simulations (1000 for each future climate scenario) is not trivial.
Section 4.2 discusses links between adelgid density and hemlock loss, but the statements are qualitative and do not seem to include quantitative components for the link, or citations for what the links are. How were these parameters defined? Is a validation data set used for this part of the model? These are important issues, and could be included as a supplemental section. Without this information, this section is difficult to assess.
RESPONSE: We added additional details and citations to section 4.2 to better explain the relationships between adelgid density and hemlock decline.
Miscellaneous and minor issues:
The legend for Figure 3 states “…for (a, c) 2015,…” should this be (a, d)?
RESPONSE: Yes, corrected.
Line 386 refers to “Confronting these models…”, the term “validating” might be suitable.
RESPONSE: The models were “tested” with the data, so it’s not really validation. We changed “Confronting” to “Testing”
In lines 64-65, and on lines 409-410, the text indicates the adelgid was introduced to eastern North America, however, this is the time when it was first documented in eastern North America. The date of its introduction is not known.
RESPONSE: As noted in response to Reviewer 1, we have changed the wording from “introduced” to “detected” or “observed” throughout.
Line 391, why is wood movement listed? While this is a path for the movement of wood boring insects (bark beetles, Emerald Ash Borer, Asian Longhorned Beetle, etc), is it a pathway for the adelgid?
RESPONSE: It is more likely that the horticultural trade is still moving the adelgid, so we have changed the wording accordingly (note now at lines 427-430).
Line 394 – 396, consider providing a table of the models to summarize their results.
RESPONSE: After careful consideration, we have opted not to include such a table. The various degrees of under- and mis-prediction of the dispersal of the adelgid would be better presented in a variety of maps. Fitzpatrick et al. (2012) [89] clearly illustrated how the models mis-predict, and it would probably add at least another page of detail, and several more figures to summarize all the mis-predictions in the literature. Finally, as the USFS hasn’t updated its own maps of the county-wide occurrence data for the adelgid in three years, it is difficult to provide current estimates of mis-specification beyond what we already illustrate in Figure 5.
Line 263, should the text “…C stocks 100 years…” be “…C stocks after 100 years…”?
RESPONSE: Corrected.
Line 44, the text indicates hemlock has been dominant, though I am not sure that is the case, perhaps “significant”, or “common”?
RESPONSE: National Vegetation Classification does list T. canadensis as a dominant or significant component. We have added “significant” (now line 49).
Line 142, consider replacing “of” with “in”
RESPONSE: Corrected.
Lines 73 – 75, consider adding citation for Morin and Liebhold 2015, Forest Ecology and Management 341:67-74.
RESPONSE: Added (now at line 76). (reference [43]).
